# Provably Robust Watermarks for Open-Source Language Models

## Abstract

Watermarking is a leading solution for the increasingly pressing problem of identifying AI-generated text. Existing large language model (LLM) watermark approaches crucially rely on the LLM's source code and parameters being secret, which makes them ineffective in an open-source setting. In this work, we introduce the first watermarking scheme for *open-source LLMs* with provable robustness guarantees. Under precisely defined assumptions about the adversary's knowledge, we prove that the adversary either fails to remove the watermark or significantly degrades the quality of the model. We supplement our theoretical results with experiments using Qwen, which show how our proven robustness-quality tradeoff manifests in practice.

Our main contribution is showing the *feasibility* of watermarks with provable guarantees in the open-source setting. We provide the first formal definition of robustness in this setting, and show that it is achievable by a fairly simple scheme. While this scheme is simple, the bulk of our work lies in modeling the problem in a way that is realistic yet amenable to provable results, and analyzing our scheme to prove robustness. We hope that our definitions and the techniques used in our analysis pave the way for future work on open-source watermarks.

## 1 Introduction

Without strong identification methods for identifying AI-generated text, we face risks such as model collapse (Shumailov et al., 2024), mass disinformation campaigns (Ryan-Mosley, 2023), and detection false positives leading to false plagiarism accusations (Ghaffary, 2023). Watermarking, where detectable patterns are embedded in model outputs, is a prominent and promising approach to detection. Recent works (Kirchenbauer et al. (2023); Aaronson (2022); Zhao et al. (2024); Christ et al. (2024); Kuditipudi et al. (2024); Fairoze et al. (2024); Christ & Gunn (2024); Golowich & Moitra (2024); Alrabiah et al. (2025)) have made significant progress in building LLM watermarks for the setting where users only have query access to the model. However, these watermarks are easily removable if an attacker has access to the model's code.

Watermarks for open source models have been severely understudied, yet open models are becoming more widely available and higher quality (e.g., LLaMA (Touvron et al., 2023), Mistral (Jiang et al., 2023), OPT (Zhang et al., 2022), Qwen Qwen Team (2026)). In this work we initiate a formal study of watermarks for open models, in the setting where the model parameters and associated code are publicly available. Constructing watermarks that are not easily removable by an attacker with such comprehensive model access is a significant challenge.

**Why sampler-based watermarks fail.** Existing LLM watermarks with provable guarantees are all "sampler-based." LLMs typically involve two components, a *sampling algorithm* and a *neural network*. To choose each token in a response, the LLM first uses the neural network to compute a vector of *logits*, which describes a probability distribution over the next token. It then uses the sampling algorithm to draw a token from this distribution. *Sampler-based watermarks* alter the sampling algorithm; for example, Zhao et al. (2024) alter the sampler to increase the probabilities of tokens on a "green list." Concretely, sampler-based watermarks are implemented by modifying the code for the sampling algorithm, and leaving the neural network unchanged. Therefore, an attacker with access to this code can easily rewrite the sampling algorithm

to be the unwatermarked version. Since the watermark left the neural network unchanged, reverting the sampling algorithm yields exactly the *original, unwatermarked model.*

**Robust sampler-based watermarks do not help us.** Those familiar with LLM watermarks may wonder how this code modification attack is possible when many sampler-based schemes come with provable *robustness* guarantees. For example, the watermark of Golowich & Moitra (2024) withstands a constant rate of edits (e.g., changing, inserting, and deleting words) to the watermarked text. However, the standard notion of robustness considers the following setting, which is subtly different than ours: an adversary queries the watermarked model *as a black box* to obtain a watermarked response $x$, and attempts to modify it to a nearby $x'$ that is unwatermarked. In our open-source setting, the adversary is given *code $C$* for a watermarked model, which includes all materials required to run the model (i.e., model weights and inference code). It then aims to produce *code $C'$* for a high-quality and unwatermarked model. An adversary's success in this code modification attack does not contradict robustness; for example, the attack that modifies the sampling code does not yield an algorithm for transforming a watermarked $x$ into a nearby unwatermarked $x'$. In fact, this attack (which applies to Golowich & Moitra (2024)) shows that there exist schemes with high "standard robustness" and no meaningful unremovability guarantees in our open-source setting.

**What one can hope for.** We've established that constructing open-source watermarks is difficult, and existing solutions are unsatisfactory. But what formal unremovability guarantee can we even hope to show? At a high level, we wish to show that it is infeasible for an adversary, given a watermarked model, to produce an unwatermarked model. However, as is the case with sampler-based watermarks, there are certain attacks that no watermarking scheme can overcome. For example, a sufficiently well-resourced adversary can simply ignore the given watermarked model, and train its own high-quality unwatermarked model from scratch. Furthermore, even an adversary with extremely limited resources can train its own model from scratch, though this model may have low quality. Therefore, any provable unremovability guarantee must take the following form: no *limited* adversary can produce a *high-quality* unwatermarked model. Our definition takes exactly this form, and we precisely define the notions of "limited" and "high-quality" that we consider.

Preventing a limited adversary from removing the watermark is still highly valuable, and this kind of limited robustness is the status quo in sampler-based watermarks. First, raising the bar of difficulty for removing increases the number of scenarios where in practice, it will not be worth the attacker's effort. By analogy, image watermarks (e.g., the Getty images watermark) can be removed with enough effort; however, they are still very useful as in many cases expending this effort is not worthwhile. Second, if removing the watermark requires degrading the quality of the content, an attacker may be deterred. Finally, some robustness is necessary for the watermark to persist even when a well-meaning user alters the model and text, not in an effort to remove the watermark.

**The key challenge.** In the open model setting, all information necessary to embed the watermark is given to the attacker. Doing so without revealing information that allows an attacker to revert the model to an unwatermarked version is challenging. For example, many watermarks can be reverted if one knows the secret watermarking key. Therefore, a watermark for open models should somehow hide the watermarking key in the weights of the watermarked model, so that no attacker can extract it. Surprisingly, we are able to do so without relying on heavy tools such as obfuscation (Barak et al., 2001), which is both impractical and not yet known to solve the problem. Instead, we rely on the adversary's uncertainty about what the last-layer weights "should look like," and plant the watermark simply by modifying these last-layer weights. Although our scheme is simple, our analysis is more involved and sheds new light on the relationship between a language model's parameters and the text it generates.

**Our contributions.** We put forth a formal definition of an *unremovable*[1] watermark for LLMs, then construct such a scheme. Our scheme carefully modifies the weights of the model's neural network, such that the adversary provably cannot reconstruct these modifications and reverse them. A key challenge is designing a strategy for modifying the weights that yields a provably detectable signal in text produced by the model. Our watermark is unremovable in the sense that, assuming the adversary has sufficient uncertainty about

---

[1]We use the term *unremovable* to refer to our technical property, and *robustness* to refer to the more general concept of withstanding modification.

the distribution of high-quality language, it cannot remove the watermark without significantly degrading the quality of the model. We prove bounds on the amount of text required to detect our watermark, and show that in practice it is detectable from $\sim 300$ tokens.

In addition to proving unremovability, we provide a suite of experiments using OPT-6.7B and OPT-1.3B, showing that our theoretical guarantees translate to practice. We find that the strongest attack we consider requires deteriorating text quality to zero out of 100 in order to bring the detection rate to 50%. Our approach is quite general, and our techniques may be of independent interest due to their applicability in other settings where one wishes to watermark high-dimensional data.

**Comparison to sampler-based watermarks.** We consider a much stronger attack model than sampler-based watermarks—before our work, no watermarks provably withstood an (even limited) attacker with access to the model parameters. As we prioritize this strong robustness requirement, the quality guarantee of our watermark is not the strongest achieved by sampler-based watermarks. For example, it is possible to construct "undetectable" sampler-based watermarks that preserve quality under any efficiently computable metric (Christ et al., 2024). In contrast, our watermark introduces noticeable, though small, distortion to the model's output distribution. This is similar to the sampler-based watermark of Zhao et al. (2024). Importantly, we show a favorable relationship between the quality degradation introduced by the watermark, and the quality degradation required to remove the watermark: the removal quality degradation is greater by a factor of nearly $\sqrt{n}$, where $n$ is the size of the token alphabet.

**Organization of the paper.** We present a high-level technical summary of the paper in Section 2. Within this section, we often refer to specific parts of the body of the paper where the corresponding full details can be found. We recommend reading the technical overview first, then reading the body of the paper (Section 3 onward), to get a sense of what to expect. However, a reader immediately interested in the full technical details can start at Section 3.

## 2 Technical Overview

**Formalizing unremovability (Full details in Section 4.1).** In this section, we present an overview of the paper. Within these overview, we refer to sections that provide the full technical details.

Recall that the goal of the adversary is to take a watermarked model and produce a high-quality unwatermarked model; we wish to capture such an adversary in our definition of unremovability. In order for unremovability to be achievable, we must consider only adversaries with limited knowledge about the distribution of high-quality text. We formalize this notion as follows: we model ideal-quality language as being described by an original neural network $M^*$. That is, the ideal-quality language is the distribution of text that would be produced by an LLM using $M^*$ to compute the probability distribution over each token. This neural network $M^*$ is then watermarked, and the resulting $M'$ is given to the adversary. The adversary's uncertainty is captured by its *posterior distribution over* $M^*$, given $M'$.

We assume that the adversary's posterior distribution over $M^*$ is normally distributed in a particular representation space of models, and centered at $M'$. Under this assumption, we prove that there is a steep trade-off between the magnitude of changes the adversary must make to remove the watermark, and the magnitude of changes required to add the watermark. That is, text produced by the adversary with access to $M'$ is either watermarked or low-quality.

**Our scheme (Full details in Section 5).** Our scheme is quite natural: We modify the bias of each neuron in the last layer of the model by adding a perturbation from $\mathcal{N}(0, \sigma^2)$, as described in Algorithms 1 and 2. The watermarking detection key is the vector of these perturbations. This key allows us to compute an "inner product score" of any model. This score is the inner product between the detection key, and the difference between the biases of the given model and the original model (Algorithm 3). So far, this inner product score is of limited use in detection because it requires the weights of the model (possibly modified by an adversary) used to produce the text being detected.

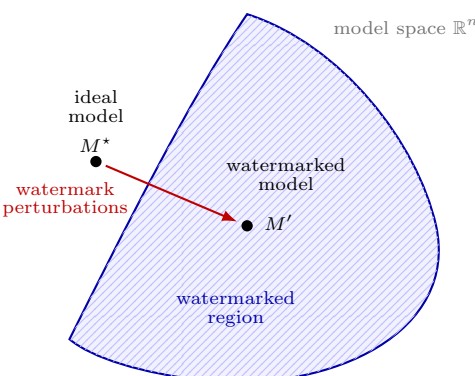

Figure 1: The unremovability game. An ideal model $M^* \in \mathbb{R}^n$ is watermarked, resulting in $M'$. The adversary receives $M'$ and attempts to alter it to obtain a vector outside of the watermarked region, but still close to $M^*$.

Our key observation is that it is possible to approximate this inner product score *given only text* produced by the model. This text detector (Algorithm 4) computes a score that is the sum of the watermark perturbations corresponding to the distinct tokens observed in the text. We show that in expectation, this score is approximately the inner product from the detector from weights (Algorithm 3), where the approximation error is lower for higher-entropy text. We prove that in sufficiently high-entropy text, our text detector succeeds with overwhelming probability.

**Proving unremovability (Full details in Theorem 3).** We show unremovability in two steps: (1) Any high-quality model produced by altering the watermarked model must have a high inner product score (Theorem 1), and (2) If a language model has a high inner product score, its responses are watermarked (Theorem 3).

To understand the intuition behind (1), consider the watermarked and unwatermarked models as vectors $\vec{w}_{\mathsf{wat}}, \vec{w}^* \in \mathbb{R}^n$, respectively. The adversary is given $\vec{w}_{\mathsf{wat}}$, and it aims to produce $z \in \mathbb{R}^n$ that is unwatermarked, but that is sufficiently close to $\vec{w}^*$ (i.e., has high quality). We can describe the region of watermarked vectors geometrically: This region is essentially a halfspace,[2] where the hyperplane describing it lies between $\vec{w}_{\mathsf{wat}}$ and $\vec{w}^*$, and it is orthogonal to the line from $\vec{w}_{\mathsf{wat}}$ to $\vec{w}^*$. Removing the watermark involves crossing this hyperplane. If $\vec{w}^*$ is known, the most efficient way to remove the watermark is to add to $\vec{w}^*$ in the direction of $(\vec{w}^* - \vec{w}_{\mathsf{wat}})$. However, because the adversary's posterior distribution over $\vec{w}^*$ is centered at $\vec{w}_{\mathsf{wat}}$, the adversary is unlikely to produce a short vector $(z - \vec{w}_{\mathsf{wat}})$ that moves far in the direction of $(\vec{w}^* - \vec{w}_{\mathsf{wat}})$: We expect its weight in this direction to be proportional to its length. This allows us to show that, if $z$ is not watermarked, then $(z - \vec{w}_{\mathsf{wat}})$ has large magnitude. If $(z - \vec{w}_{\mathsf{wat}})$ has large magnitude, then $z$ is far from $\vec{w}^*$ and therefore low quality.

Now consider some text produced by a model on the watermarked side of the hyperplane. For each token, if the watermark increases its bias relative to that of $\vec{w}^*$, the model is more likely to output that token. Prior work of Zhao et al. (2024) uses this principle to obtain a simple "counting" detector, which essentially computes the fraction of positive-bias tokens in the given response. However, we observe that because our watermark alters different token biases by different amounts, we can substantially improve detectability by taking the magnitudes of the perturbations into account in our detector. Our detector computes a score, which is the sum of bias perturbations of all observed tokens in the text (in contrast to the counting detector, which is the sum of indicators of the bias perturbations' *signs*). Interestingly, we prove that our text detector, when applied to text produced by $z$, approximates the inner product between $(z - \vec{w}^*)$ and $(\vec{w}_{\mathsf{wat}} - \vec{w}^*)$. Because this inner product is high for watermarked $z$ and roughly 0 for independently generated $z$, our detector is reliable and has a negligible false positive rate. Similarly to other watermarks (e.g., Kirchenbauer

---

[2]More precisely, we define it to be the intersection of a halfspace and a ball, to ensure a low false positive rate.

et al. (2023); Christ et al. (2024); Zhao et al. (2024)) our proof of detectability from text (2) relies on sufficient entropy in the text.

When proving unremovability of the text watermark (2), we assume that the adversary alters only the biases in the last layer of the watermarked model it is given. We further discuss this assumption and our other modeling choices in Section 2.1. We argue that an adversary successfully making arbitrary alterations to the model could use this new model to learn ways to alter only the biases of the watermarked model. We leave a formal expansion of the class of adversaries to future work.

**Technical assumptions and experiments (Full details in Section 6).** Our proof of unremovability uses several technical modeling assumptions, stated precisely in Section 5.2. For example, Assumption 2 states that two functions of the model's logits are approximately equal. In addition to justifying these assumptions with explanations in Section 5.2, we provide experiments to show that our watermark's performance in practice follows the theory we develop. This reinforces that our technical assumptions hold for typical text.

We show that our watermark is indeed detectable and unremovable in practice, using experiments run on Qwen3.5-4B. We use Mistral-7B Instruct (Jiang et al., 2023) to measure the quality of responses, and we show that the watermarked models with our scheme preserve similar quality to their unwatermarked version. We demonstrate reliable detectability rates for our construction, given ∼300-token responses of various types. In addition to our proofs of unremovability, we consider a number of concrete adversaries and demonstrate that the watermark persists in the adversarially altered watermarked models unless the adversary adds large perturbations that significantly reduce the quality of the resulting model. In line with our definition of unremovability, we consider adversaries with limited knowledge of natural language, therefore considering concrete attacks that do not require significant amounts of data (unlike, e.g., fine-tuning attacks).

## 2.1 Our modeling choices

The problem we consider requires us to work with many concepts that are difficult to define formally, such as the quality of text. As a result, we make several modeling choices, which we discuss here.

**Our quality notion.** We use the Euclidean distance between model weights as a proxy for the quality of a model. Evaluating the quality of a model in practice is challenging, usually involving a huge suite of expensive experiments. As a result, when defining a quality metric it is difficult to balance fidelity to practical quality, with the feasibility of proving results with respect to this metric. We emphasize that the main importance of our quality metric in our results, is showing the amount of quality degradation the attacker introduces, relative to the amount of degradation introduced by the watermark itself. We show that the attacker's degradation is greater by a large fraction (of roughly $\sqrt{n}$), which we expect to outweigh imprecision in the quality metric in general.

One naturally wonders how this Euclidean distance quality metric translates to a more intuitive notion of text quality. We study this using experiments, where we implement attackers altering the weights by varying Euclidean distance, generate responses from these attacked models, and ask an outside model to evaluate their quality. This LLM-as-a-judge quality evaluation follows prior work Piet et al. (2023). Our experiments reinforce that Euclidean distance indeed appears to translate to this practical notion of quality. However, we emphasize that this LLM-as-a-judge quality metric should be interpreted as a coarse text-quality proxy, rather than an assurance that the model performs well across all tasks. We leave a comprehensive empirical study of open-source watermarks to future work.

**Our adversary.** We consider an adversary that changes only the last layer biases of the model, and leaves the other weights unchanged. This is the most significant limitation of our robustness guarantee. In practice, we hope that changing the other weights would compromise model quality even more, unless the attacker has significant resources (e.g., a training dataset of high-quality text). Furthermore, we suspect that an attacker that changes all of the weights may be used to construct an attacker that changes only the last layer weights. That is, suppose an attacker constructs an (arbitrarily different) neural network that is high-quality and unwatermarked. This means that the frequencies of the tokens output by this network contain little signal from the watermark perturbations. By comparing the token frequencies of the watermarked model

with the token frequencies under this attacker's model, one should be able to learn which tokens' biases were increased or decreased by the watermark. This information would allow modifying the last-layer biases to reverse the watermark perturbations. We leave formalizing this intuition for future work.

**Robustness to practical attacks.** We focus on provable robustness guarantees, and emphasize that the fact that robustness for an open-source watermark can be proven at all is a significant step forward. Our unremovability guarantee applies to a large class of adversaries but does not capture *all* practical attacks, namely as those using significant knowledge about the distribution of high-quality text (e.g., fine-tuning attacks, which require data). Analogously, many sampler-based watermarks are provably robust to certain classes of attacks such as a bounded fraction of token substitutions, but do not resist all practical attacks such as paraphrasing. There, research aims to broaden the class of resisted attacks, while recognizing that robustness to arbitrary attacks is too lofty a goal; we have the same aim here. We are the first to prove robustness to *any* class of attacks in the open-source setting.

**Robustness to post-edits.** We focus on robustness to an adversary that attempts to create an entire unwatermarked model. In a slightly different (and more standard) setting, an adversary may use the watermarked model to create a response, then edit that response to remove the watermark. Our watermark is similar to that of Zhao et al. (2024), which has robustness to a bounded fraction of word changes, and we show in preliminary experiments that our scheme displays some substitution-robustness (Figure 5). However, post-edit robustness is not the focus of this work, and we defer a full study to future work.

**Model structure.** In our theoretical analysis, we consider only models with last-layer biases. While this structure is standard, there do exist models with other structures. An interesting direction for future work is designing watermarks for open models with different architectures. We also note that if an open model provider wishes to watermark their model, they should just use the last-layer bias architecture.

## 3 Related work

We describe the most relevant related works here, but refer readers to (Zhao et al., 2025; Tang et al., 2023; Piet et al., 2023) for more comprehensive surveys.

Existing watermarks change the sampling function of the LLM, and are therefore easily removable given code for this sampling function. Aaronson (2022); Kirchenbauer et al. (2023); Zhao et al. (2024); Christ et al. (2024); Fairoze et al. (2024) sample each token by partitioning the token set into red and green lists (that depends on the previously-output tokens), then increasing the probability of the tokens in the green list. To detect the watermark, one computes the fraction of green tokens in the given text and compares it to a threshold. Kuditipudi et al. (2024); Christ & Gunn (2024) operate slightly differently—for each response, they choose a random seed. In Kuditipudi et al. (2024) there are polynomially many possible seeds for a given watermarking key, and in Christ & Gunn (2024) there are exponentially many. They then choose the $i^{\text{th}}$ token of the response to be correlated with the $i^{\text{th}}$ value of the random seed. To detect the watermark, one essentially computes the correlation between the response and the seed. While not unremovable, some of these watermarks satisfy a weaker *robustness* property: Given a watermarked response, it is difficult to make a bounded number of edits to yield an unwatermarked text.

Gloaguen et al. (2025) identify desirable properties of open-source watermarks (e.g., *durability*) and show that existing sampler-based watermarks do not satisfy these properties; however, they do not provide a solution.

Our scheme is most similar in spirit to that of Zhao et al. (2024), though theirs is a sampler-based watermark. They choose a fixed red/green partition that is used to change the sampling function for all responses. Similarly, we fix the partition of tokens whose biases we increase and decrease. The two major differences are that (1) we make this change by altering the weights of the model, and (2) while the scheme of Zhao et al. (2024) increases the logits of all green tokens by the same amount, our increases are normally distributed. This normal distribution is crucial for unremovability, and it results in a different detector being most effective for our scheme. The detector of Zhao et al. (2024) simply computes the fraction of green tokens, while our

inner product detector computes a score that is essentially a weighted count of the number of green tokens. These weights correspond to the size of the perturbations added by our watermark.

Block et al. (2025), in a work released after ours, use our strategy of adding Gaussian perturbations to the model weights to embed a watermark. However, they do not consider the open-source setting or analyze robustness to an adversary that knows the model's weights.

The works closest to our setting are Gu et al. (2024); Sander et al. (2024), which show empirically that certain sampler based watermarks can be learned. That is, a model trained on watermarked texts may learn to generate watermarked responses itself. The watermark therefore is embedded in the weights of the model; however, there is no unremovability guarantee, and the detectability of these responses is much lower than their sampler-based counterparts. In fact, Gu et al. (2024) find empirically that the watermark is destroyed after the model is fine-tuned and leaves the question of unremovable open-source watermarks for future work.

Zhang et al. (2024) show that any LLM watermark is removable by a sufficiently strong adversary. This underscores our need to consider an adversary with limited knowledge about high-quality language when defining and showing unremovability.

## 4 Preliminaries and definitions

Let $\mathbb{N} := \{1, 2, \ldots\}$ denote the set of positive integers. We will write $[q] := \{1, \ldots, q\}$. For a set $X$, we define $X^* := \{(x_1, \ldots, x_k) \mid x_1, \ldots, x_k \in X \land k \in \mathbb{Z}_{\geq 0}\}$ to be the set of all strings with alphabet $X$. For a binary string $s \in X^*$, we let $s_i$ denote the $i^{\text{th}}$ symbol of $s$ and $\text{len} s$ denote the length of $s$. For a string $s \in X^*$ and positive integers $a \leq b \leq \text{len} s$, let $s[a : b]$ denote the substring $(s_a, \ldots, s_b)$. We use $\log(x)$ to denote the logarithm base 2 of $x$, and $\ln(x)$ to denote the natural logarithm of $x$. For a finite set $X$, we will use the notation $x \leftarrow X$ to denote a uniformly random sample $x$ from $X$. If $X$ is a set of $n$-dimensional column vectors, we will write $X^m$ to refer to the set of $n \times m$ matrices whose columns take values in $X$. Unless otherwise specified, vectors are assumed to be column vectors. For a vector $x \in \mathbb{R}^n$, we let $\|x\| = \|x\|_2 = \sqrt{\sum_{i=1}^n x_i^2}$.

Let $\text{Ber}(p)$ be the Bernoulli distribution on $\{0, 1\}$ with expectation $p$. Let $\text{Ber}(n, p)$ be the distribution on $n$-bit strings where each bit is an i.i.d sample from $\text{Ber}(p)$. For a distribution $\mathcal{D}$, we let $\text{Supp}(\mathcal{D})$ denote its support.

We use notation and notions that are standard in the cryptography literature but may be less familiar to other readers. We let $\lambda$ denote the *security parameter*, which typically parameterizes how difficult it is for an attacker to break the security of a scheme. We wish that bad events happen with probability that is *negligible*, or decreasing very quickly with $\lambda$. More precisely, a function $f$ of $\lambda$ is *negligible* if $f(\lambda) = O(\frac{1}{\text{poly}(\lambda)})$ for every polynomial $\text{poly}(\cdot)$. We write $f(\lambda) \leq \text{negl}(\lambda)$ to mean that $f$ is negligible. We typically want good events to occur with probability approaching 1 quickly as $\lambda$ grows; we say a probability is *overwhelming* in $\lambda$ if it is equal to $1 - f$ for some negligible function $f$. We let $\approx$ denote computational indistinguishability and $\equiv$ denote statistical indistinguishability.

We provide a formal description of a language model in Section B.

### 4.1 Watermarks

We present definitions for watermarking model weights from a distribution $\mathcal{C}$ of real vectors; that is, $\text{Supp}(\mathcal{C}) \subseteq \mathbb{R}^n$.

**Definition 1** (Watermark). *A watermark is a tuple of polynomial-time algorithms $\mathcal{W} = (\text{Setup}, \text{Watermark}, \text{Detect})$ such that:*

- $\text{Setup}(1^\lambda) \rightarrow \text{sk}$ *outputs a secret key, with respect to a security parameter $\lambda$.*

- $\text{Watermark}_{\text{sk}}(x) \rightarrow x_{\text{wat}}$ *is a randomized algorithm that takes as input weights $x$ and outputs watermarked weights $x_{\text{wat}}$.*

- $\mathsf{Detect_{sk}}(x') \rightarrow \{\mathsf{true}, \mathsf{false}\}$ *is an algorithm that takes as input weights $x'$ and outputs* $\mathsf{true}$ *or* $\mathsf{false}$.

The input $1^\lambda$ is standard notation in cryptography, meaning a unary $\lambda$-length string. This simply enforces that the algorithm in question runs in time polynomial in $\lambda$.

**Definition 2** (Quality loss function). *A quality loss function is a function $L : \mathbb{R}^n \times \mathbb{R}^n \rightarrow \mathbb{R}$. We say that $L(x, y)$ is the* quality loss score *of the weights $y \in \mathbb{R}^n$ relative to some ideal weights $x$.*

The following unremovability game is defined over a model weight distribution $\mathcal{C}$, a quality measure $L$ over $\mathbb{R}^n$, and a loss parameter $\ell(\cdot) : \mathbb{Z} \rightarrow \mathbb{R}$. The quality measure $L$, for example Euclidean distance, specifies the quality notion we consider. The loss parameter $\ell$ determines how much quality loss is acceptable, as a function of the model size $n$.

**Definition 3** (($\mathcal{C}, L, \ell$)-Unremovability game $\mathcal{G}_{\mathcal{A}, \mathcal{W}, \mathcal{C}}^{remov}(1^\lambda, L, \ell)$). *Let $\mathcal{C}$ be a weight distribution. The unremovability game $\mathcal{G}_{\mathcal{A}, \mathcal{W}, \mathcal{C}}^{remov}(1^\lambda, L, \ell)$ is defined between an adversary $\mathcal{A}$ and a challenger as follows.*

1. *The challenger runs $\mathsf{sk} \leftarrow \mathsf{Setup}(1^\lambda)$.*

2. *The challenger chooses some original weights $x \leftarrow \mathcal{C}$ and produces watermarked weights $x_{\mathsf{wat}} \leftarrow \mathsf{Watermark_{sk}}(x)$.*

3. *$\mathcal{A}$ receives $x_{\mathsf{wat}}$ and produces $x'$.*

4. *The adversary wins if $x'$ is not watermarked, and its quality loss is acceptable: $L(x, x') \leq \ell(n)$.*

*The output of the game $\mathcal{G}_{\mathcal{A}, \mathcal{W}, \mathcal{C}}^{remov}(1^\lambda, \mathcal{C}, L, \ell)$ is 1 if and only if the adversary wins.*

**Definition 4** (Unremovability). *Let $\mathcal{W}$ be a watermark for weights in $\mathbb{R}^n$. Let $\mathcal{C}$ be a weight distribution, $L$ be a quality loss function over $\mathbb{R}^n$, and $\ell(\cdot) : \mathbb{R}^n \times \mathbb{R}^n \rightarrow \mathbb{R}$ be a loss parameter. We say that $\mathcal{W}$ is $(\mathcal{C}, L, \ell)$-unremovable if for all p.p.t. adversaries $\mathcal{A}$, $\Pr\left[\mathcal{G}_{\mathcal{A}, \mathcal{W}, \mathcal{C}}^{remov}(1^\lambda, L, \ell) = 1\right] \leq \mathsf{negl}(\lambda)$.*

**Definition 5** (Soundness/low false positive rate). *Let $\mathcal{W}$ be a watermark for weights in $\mathbb{R}^n$. $\mathcal{W}$ is* sound *if for any fixed $x \in \mathbb{R}^n$, $\Pr_{\mathsf{sk} \leftarrow \mathsf{Setup}(1^\lambda)}\left[\mathsf{Detect_{sk}}(x) = \mathsf{true}\right] \leq \mathsf{negl}(\lambda)$.*

As shorthand, when the setup and watermark algorithms are clear from context, we sometimes write that $\mathsf{Detect}$ (rather than $\mathcal{W}$) is unremovable or sound.

# 5 Watermarking scheme

For presentation's sake, we first introduce a general watermarking scheme for vectors in $\mathbb{R}^n$, with a detector $\mathsf{WeightDetect}$. This is not meant as a standalone scheme; rather it is useful in proving unremovability of our end scheme. We show that if this general scheme is applied to the last-layer biases of a neural network, any high-quality model produced by an adversary in our unremovability game is still watermarked. That is, the last-layer biases in the adversary's model are still detected by $\mathsf{WeightDetect}$. We then introduce $\mathsf{TextDetect}$, our final detector, which uses only text to determine the presence of a watermark. We show that for language models, if the last-layer biases are detected by $\mathsf{WeightDetect}$, the text generated by the model is detected under $\mathsf{TextDetect}$. This requires careful analysis of how the last-layer biases affect the distribution of tokens output by the model (Section 5.2).

## 5.1 Measuring watermark signal in the weights of the model

We consider an arbitrary vector $\vec{w}^*$ in $\mathbb{R}^n$ to which Gaussian noise is added, to obtain $\vec{w}_{\mathsf{wat}}$. Observe that the inner product between $(\vec{w}_{\mathsf{wat}} - \vec{w}^*)$ and the vector of the Gaussian perturbations $\Delta$ is large. Naturally, our detector $\mathsf{WeightDetect}$ (Algorithm 3) computes this inner product. We show that any adversary whose posterior distribution (after seeing $\vec{w}_{\mathsf{wat}}$) over $\vec{w}^*$ is Gaussian cannot remove the watermark without adding significant perturbations of its own. In particular, if $\vec{w}^*$ has dimension $n$, the adversary must add a vector of Euclidean norm $\Omega(n/\sqrt{\log n})$ to succeed at removing the watermark. In contrast, embedding the watermark

---

**Algorithm 1:** Watermark setup algorithm Setup

---
**Result:** Watermark secret key $\Delta \in \mathbb{R}^n$
**return** $\Delta \leftarrow \mathcal{N}(0, \sigma^2 I)$;

---

**Algorithm 2:** Watermarked weight generator Watermark

---
**Input:** Content $x \in \mathbb{R}^n$ (later taken to be the last-layer biases of a model) and watermark secret key
$\qquad \Delta \in \mathbb{R}^n$
**Result:** Watermarked content $x' \in \mathbb{R}^n$ and original content $x$
$x' \leftarrow x + \Delta$;
**return** $x', x$;

---

**Algorithm 3:** Detector WeightDetect$_{\Delta, \tau}$

---
**Input:** Content $c \in \mathbb{R}^n$, original content
$\qquad x \in \mathbb{R}^n$, key $\Delta \in \mathbb{R}^n$, and detection
$\qquad$ threshold $\tau$
**Result:** true or false depending on whether $c$ is
$\qquad$ watermarked
**if** $(c - x) \cdot \Delta \geq \tau \sigma^2 n$ and $\|c - (x + \Delta)\| \leq \frac{1}{2} \sigma^2 n$
**then**
$\qquad$ **return** true;
**return** false;

---

**Algorithm 4:** Detector TextDetect$_{\Delta, \tau_{\text{text}}}$

---
**Input:** Text $x_1, \dots, x_\ell \in \mathcal{T}$, detection key
$\qquad \Delta \in \mathbb{R}^n$, and detection threshold $\tau_{\text{text}}$
**Result:** true or false
$S \leftarrow \emptyset$; count $\leftarrow 0$;
**for** $i \in [\ell]$ **do**
$\qquad$ **if** $x_i \notin S$ **then**
$\qquad\qquad$ count $\leftarrow$ count $+ \Delta(x_i)$;
$\qquad\qquad$ $S \leftarrow S \cup \{x_i\}$;
$\qquad$ **if** $|S| \geq \lambda \wedge$ count $\geq |S| \sigma^2 \tau_{\text{text}}$ **then**
$\qquad\qquad$ **return** true;
**return** false;

---

Figure 2: The algorithms $\mathcal{W} = (\text{Setup}, \text{Watermark}, \text{WeightDetect}, \text{TextDetect})$.

involves adding a perturbation of Euclidean norm only $O(\sqrt{n})$: Watermark removal requires a change that is larger than watermark embedding *by a factor of nearly $\sqrt{n}$*. So far, this approach is general to watermarking any real vector. Although our work focuses on applying it to LLMs, it is likely that this paradigm can be applied to other scenarios, which we leave for future work.

To apply our watermark to LLMs, we will later take $\vec{w}^*$ to be the biases in the last layer of a neural network, making $n$ the size of the token alphabet.

**Definition 6** ($L_2$: Euclidean quality loss function.)**.** *Let $L_2 : \mathbb{R}^n \times \mathbb{R}^n \to \mathbb{R}$ be the quality loss function that, on input $x, y \in \mathbb{R}^n$, outputs $\|x - y\|_2$. We call $L_2$ the* Euclidean loss function.

**Theorem 1.** *Let $I$ be the $n \times n$ identity matrix, let $\tau \in (0, 1)$ be a constant, and let $\mathcal{C}$ be such that the adversary's posterior distribution over the original content $\vec{w}^*$ after seeing $\vec{w}_{\text{wat}}$ is $\mathcal{N}(0, \sigma^2 I)$.*

*Let the loss parameter be $\ell(n) := \frac{\sigma n}{\sqrt{\log n}}$. Then WeightDetect is $(\mathcal{C}, L_2, \ell)$-unremovable.*

**Theorem 2.** WeightDetect *is sound.*

We defer the proofs of Theorems 1 and 2 to Sections C.1 and C.2 respectively.

## 5.2 Detecting the watermark from text

In this section, we show that the inner product from WeightDetect can be approximated given text. Our detector from text, TextDetect (Algorithm 4), simply computes the sum of the perturbations of the biases of tokens in the given text. A positive perturbation increases the probability of outputting the given token; the larger the perturbation, the larger the increase. Similarly, a negative perturbation decreases a token's likelihood. Therefore, we expect the frequency of a token in a watermarked response to have an observable correlation with the sign and magnitude of the perturbation added to its bias.

We prove in Theorem 3 that this intuition is indeed correct. In particular, this sum of perturbations of observed tokens approximates a value that is 0 in expectation for natural text, and grows linearly with the text length for watermarked text. This is true even when the watermarked model is modified by an adversary attempting to remove the watermark. The accuracy of the detector's approximation depends on the entropy of the text, and the quality of the model produced by the adversary. That is, low-entropy responses and low-quality adversarial models will have lower watermark detectability.

We first introduce some notation and recall the structure of a language model. Let $z$ be the biases of the model produced by an adversary modifying the watermarked model $\vec{w}_{\text{wat}}$. Let $p_t^i$ and $q_t^i$ be the probabilities that the models $z$ and $\vec{w}^*$, respectively, assign to token $t \in [n]$ at step $i$. Let $\ell_r^i$ be the logit of token $r$ in step $i$, under the original model $\vec{w}^*$. Therefore, $p_t^i = \frac{\exp(\ell_t^i + z_t)}{\sum_{r \in [n]} \exp(\ell_r^i + z_r)}$ and $q_t^i = \frac{\exp(\ell_t^i + \vec{w}_t^*)}{\sum_{r \in [n]} \exp(\ell_r^i + \vec{w}_r^*)}$. Note that throughout our theoretical analysis, we take the temperature to be 1.

Recall that language models compute their probability distribution using the softmax function. Therefore, no probabilities are actually zero but are rather extremely small values, which are practically zero. We assume there is some threshold for which probabilities above that threshold are *significant*, and probabilities below that threshold are functionally zero—in practice, those tokens are not sampled. We treat these negligible probabilities as zero. We will use the following assumptions:

**Assumption 1** (Similar normalization). $\sum_{r \in [n]} \exp(\ell_r^i + z_r) \approx \sum_{r \in [n]} \exp(\ell_r^i + \vec{w}_r^*)$.

**Assumption 2** (Derived from similar normalization, with concrete approximation error). *There exists a very small constant $\eta$ such that for every $t \in [n]$, $p_t^i/q_t^i - 1 - \eta \leq (z_t - \vec{w}_t^*) \leq p_t^i/q_t^i - 1 + \eta$.*

Assumption 1 states that the under the adversary's model and the original model, the normalization factors used in the softmax function are roughly equal. Intuitively, for an attack that significantly changes this normalization factor, there is a similar attack that makes smaller renormalized changes. For example, if the adversary increases all biases by a large amount, it might as well change them by a smaller amount with the same ratios between them. Furthermore, the quality requirement bounds the adversary's changes $z_r$. Assumption 2 is derived from Assumption 1 and an additional approximation that $e^x \approx 1 + x$. We make this approximation error concrete by introducing $\eta$, which we assume to be small. We show this derivation in the proof of Lemma 1.

**Definition 7** ($c_1$-high min-entropy). *Let $T \subseteq [n]$ be the subset of tokens over which $p^i$ has significant probability. We say that $p^i$ has $c_1$-high min-entropy for $c_1 > 0$ if for all $t \in T$, $p_t^i \geq \frac{1}{c_1|T|}$.*

In other words, Definition 7 states that for tokens with significant probability under $p^i$, their probabilities are within a constant factor of $1/|T|$. That is, $p^i$ is somewhat close to uniform over $T$.

**Definition 8** ($c_2$-high quality). *Let $T \subseteq [n]$ be the subset of tokens over which $p^i$ has significant probability. We say that $p^i$ has $c_2$-high quality for $c_2 > 0$ if for all $t \in T$, $q_t^i \geq p_t^i c_2$.*

$c_2$-high quality states that the probabilities under the original model are within a constant factor of the probabilities under the adversary's model. This is true if the adversary makes bounded changes—recall that $p_t^i \approx q_t^i \cdot \exp(z_t - \vec{w}_t^*)$. Therefore, it is true if $(z_t - \vec{w}_t^*) \leq -\log c_2$.

Observe that $c_1$-high min-entropy and $c_2$-high quality together imply the following fact:

**Fact 1.** *Let $p^i$ be $c_1$-high min-entropy and $c_2$-high quality, and let $T$ be the set of tokens with non-negligible probability under $p^i$. Then for all $t \in T$, $1/q_t^i \leq \frac{c_1|T|}{c_2}$.*

**Lemma 1.** *Let $x_1, \ldots, x_k$ be any sequence of tokens generated using a model produced by the adversary. For all $i \in [k]$ with $c_1$-high min-entropy (Definition 7) and $c_2$-high quality (Definition 8), under reasonable assumptions (Assumptions 1 and 2), we have that $\mathbb{E}[(\vec{w}_{\text{wat}} - \vec{w}^*)_{x_i}] \geq c_1 \sigma^2/c_2 - \alpha$, where $\alpha = \frac{c_2 \eta \sigma \sqrt{2/\pi}}{c_1}$ is a very small approximation error term.*

**Assumption 3** (Independence). *Consider a response output by the adversary's model, and let $x_1, \ldots, x_k$ be a subsequence of distinct tokens in this response. The random variables $(\vec{w}_{\text{wat}} - \vec{w}^*)_{x_i}$ are independent.*

This assumption is concerned with a degenerate case where the fact that a model output tokens with positive perturbation (e.g., $(\vec{w}_{\text{wat}} - \vec{w}^*)_{x_i} > 0$) makes it less likely to output positive-perturbation tokens in the future.

Note that this assumption is about the *differences in biases between the watermarked model and original model.* Although the model's choices of tokens themselves are far from independent, these *bias differences* should be. Intuitively, our assumption about the adversary's posterior distribution says it does not know which tokens had their biases increased versus decreased. Therefore, from its perspective, for any token sequence it generates, the bias perturbations of these tokens should be uncorrelated.

This is very similar to an assumption used in (Kirchenbauer et al., 2023, Lemma E.1) that the green list is freshly drawn for each token in the response; and an assumption used in Zhao et al. (2024) called "homophily". Despite these assumptions being unproven, empirically these watermarks appear to be detectable. Our experiments, presented in Section 6, provide further evidence for our assumption, showing that our watermark is also detectable in practice.

Our main theorem states that if an adversary is given a watermarked model and it arbitrarily modifies its biases, the resulting model still produces watermarked text with overwhelming probability. This probability increases as $\sigma$ and the number of distinct tokens increase.

**Theorem 3.** *When $\mathcal{W}$ is applied to the last-layer biases of a model,* TextDetect *detects the watermark given an adversarially produced text with enough distinct tokens, enough entropy, and high enough quality.*

*More precisely, let $I$ be the $n \times n$ identity matrix, and let $\mathcal{C}$ be such that the adversary's posterior distribution over the original model $x$ after seeing $x_{\mathsf{wat}}$ is $\mathcal{N}(0, \sigma^2 I)$.*

*Let the adversary produce a model and a response generated by that model such that:*

- *The response contains at least $\lambda$ distinct tokens.*
- *At least a constant $\delta > 0$ fraction of the distributions $p^i$ of these distinct tokens are $c_1$-high min-entropy and $c_2$-high quality, for constants $c_1, c_2 > 0$. We let $\alpha = \frac{c_2 \eta \sigma \sqrt{2/\pi}}{c_1}$ be the corresponding approximation error term.*

*Under reasonable assumptions (Assumptions 1 to 3),* TextDetect *with threshold $\tau_{text} = \delta c_2/2c_1 - \alpha/2$ outputs* true *with overwhelming probability.*

**Theorem 4.** *For any constant threshold $\tau_{text} > 0$,* TextDetect *is sound.*

We defer the proof of Lemma 1 to Section C.3, and the proofs of Theorems 3 and 4 to Sections C.4 and C.5 respectively.

## 6 Experiments

In this section we present the results of an experimental evaluation of our watermarking construction which demonstrate its behavior under concrete parameter settings. Our experiments aim to show *feasibility* rather than optimality, as we open the door to further open-source watermarking work with experimental findings consistent with our theoretical results.

**Experimental Setup.** We watermark Qwen3.5-4B Qwen Team (2026). In all of our plots, we compute the fraction of responses detected, for a range of $\sigma$ parameter values. Recall that $\sigma$ is the standard deviation of the Gaussian perturbations added to the biases. Therefore, the detectability of the watermark increases with sigma.

We generate responses of up to 300 tokens using random sampling, at a temperature of 0.9. We set the maximum ngram repeat length to 5, to avoid overly repetitive responses. We generate three types of responses: story responses, essay responses, and code responses. We use the following prompts:

**Story prompt:** "Here is one of my favorite stories: It was a "

**Essay prompt:** "Here is one of my favorite essays: It is often thought that "

**Code prompt:** "Here is a python script for your desired functionality: import "

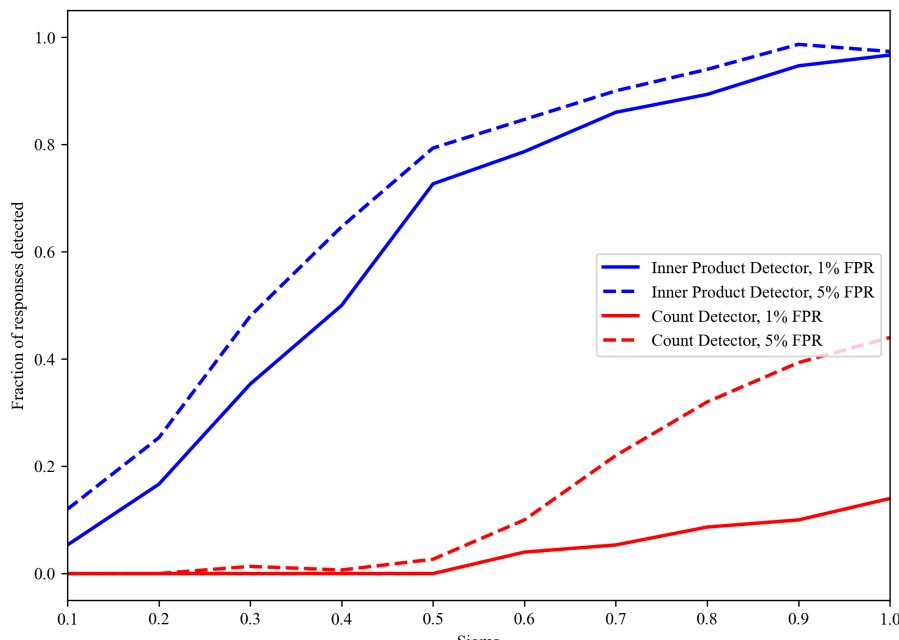

Figure 3: True positive detection rates for responses generated by Qwen3.5-4B, with our watermark applied under varying perturbation magnitudes (sigma). The "Inner Product Detector" is our TextDetect detector (Algorithm 4), and the "Count Detector" is a baseline detector that we compare to.

To evaluate quality, we ask Mistral-7B-Instruct to evaluate each response based on its type, and to provide a score out of 100.

**Detectability.** In Figure 3, we plot the fraction of watermarked responses that are detected at 1% and 5% false positive rates. These responses are output by a model watermarked using our scheme, without further modification. The curves for "Inner Product Detector" show detectability under our detector TextDetect from Algorithm 4. Observe that for $\sigma \geq 0.6$, we have a true positive rate of at least 80%. Furthermore, this plot supports our result (Theorem 3) that the true positive rate increases with $\sigma$. We emphasize that although our detectability is not as strong as some inference-time watermarks, ours is the only *open-source* watermark with a provable unremovability guarantee.

We also plot detectability for an alternate detector ("Count Detector"). This alternate detector is analogous to that of Kirchenbauer et al. (2023); Zhao et al. (2024) in that it computes the fraction of tokens whose perturbations were positive (e.g., green list tokens). Observe that this detector has a significantly lower true positive rate than of TextDetect for our scheme. This shows that the open-source setting requires new techniques for reliable detection, and that existing inference-time detectors do not carry over even when implementable in the weights of the model. This supports our strategy of considering the magnitudes of the perturbations during detection, in our Inner Product Detector.

**Unremovability.** While we prove the unremovability properties of our watermark in Section C.4, in this section we consider a concrete attack adversary and show the unremovability-quality trade-off for it. The adversary produces a new model by adding Gaussian perturbations of mean zero and standard deviation 1x sigma, 2x sigma, or 5x sigma. In Figure 4, we plot detection rates and quality scores for responses generated by adversarially modified watermarked models. That is, a watermarked model is produced at the given parameter sigma shown on the x-axis.

In Figure 4, we observe that in the 1x attack, for $\sigma \geq 0.6$ the detection rate remains above 80%. The 5x attack, shown in red, has nearly zero quality for values of sigma at least 0.6. The 1x and 2x attacks are still

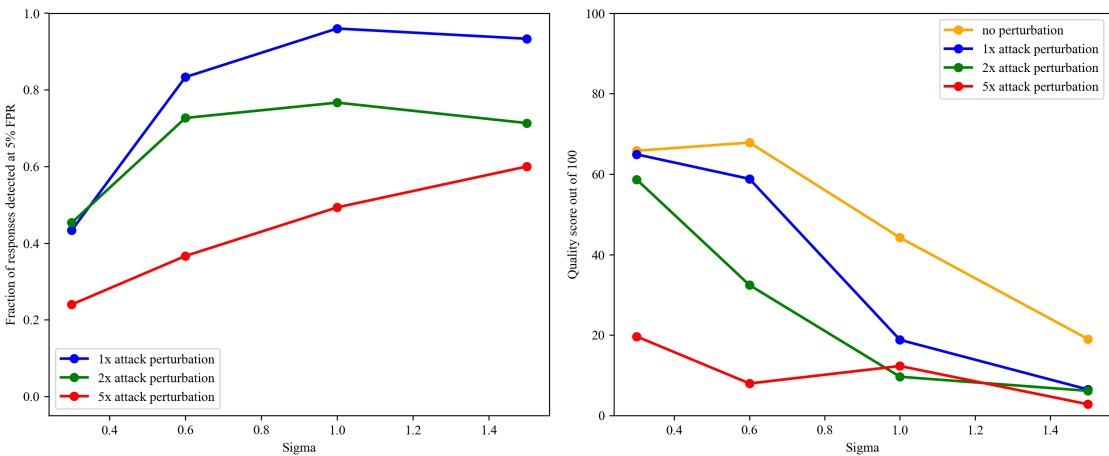

Figure 4: Left: Detection rates for adversarially perturbed watermarked Qwen3.5-4B models, to simulate a removal attack. The x-axis plots the sigma parameter used in the watermark. The three curves show detection rates for models obtained by adding additional Gaussian noise to the biases, of standard deviation 1, 2, and 5 times sigma.
Right: Quality scores of watermarked texts generated with various parameters of sigma, under the same perturbation attacks.

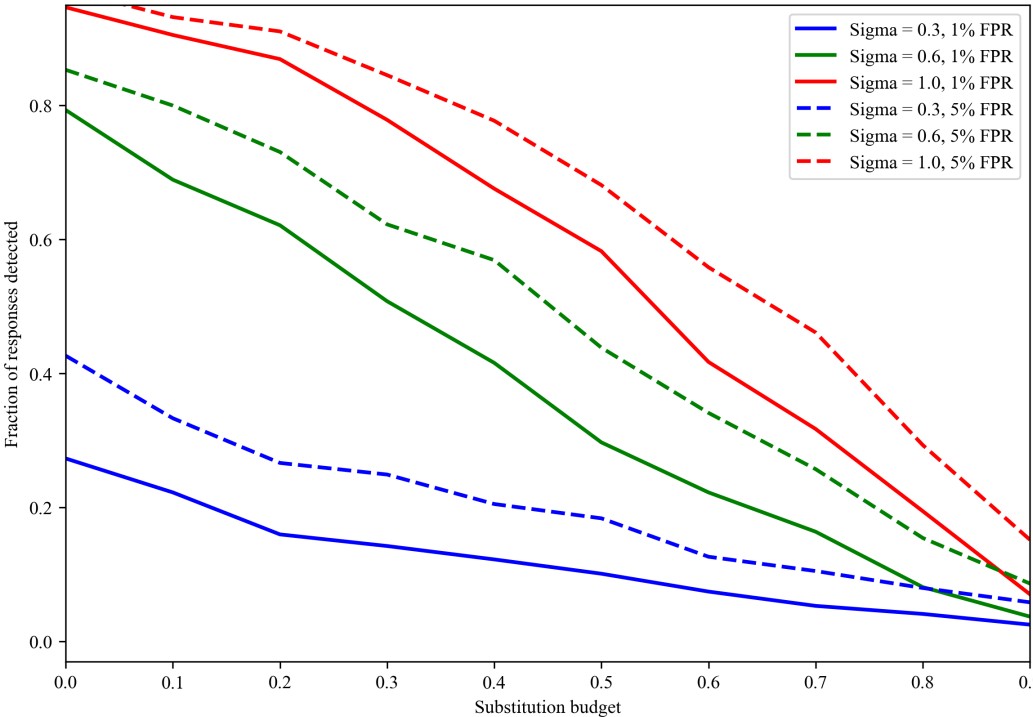

Figure 5: Detection rates for texts produced by a watermarked Qwen3.5-4B model and subjected to a substitution attack. The x-axis plots the fraction of tokens that are substituted. The curves show detection rates for varying sigma parameters of the watermark.

detectable with at least 70% probability at $\sigma = 1$, at which point the quality significantly drops to roughly 20.

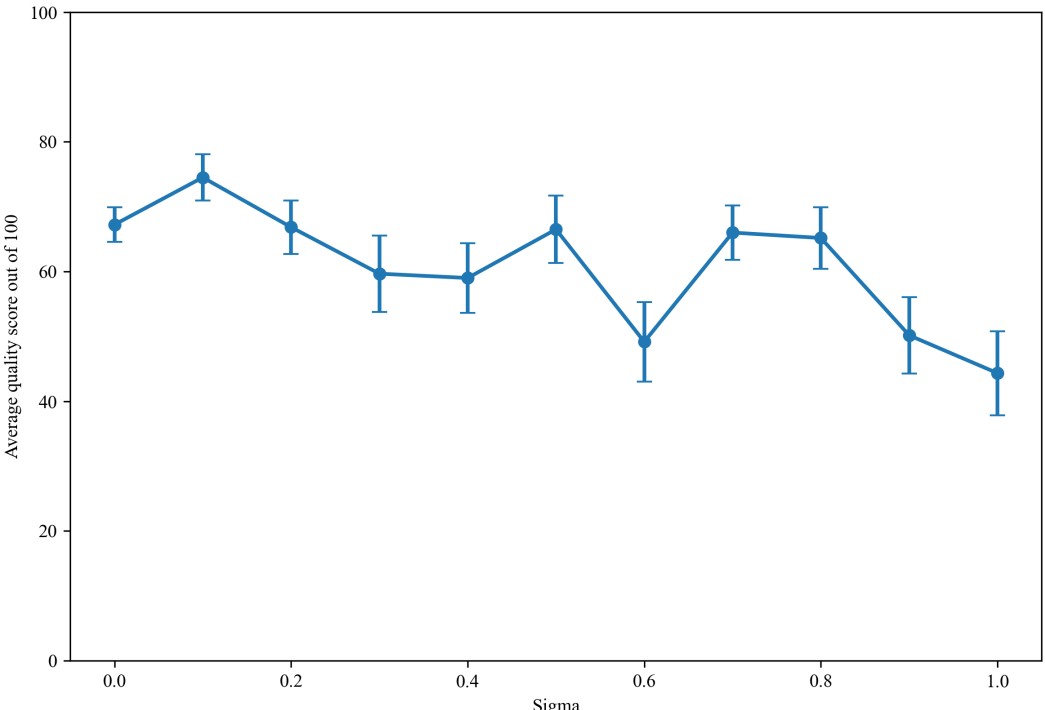

Figure 6: Quality scores, with standard error bars, of watermarked texts generated with various values of sigma.

In Figure 5, we show detection rates for watermarked responses subjected to a token substitution attack, as considered in Kuditipudi et al. (2024). We reproduce their attack, choosing a random subset of tokens in the response to substitute with uniform tokens from the token alphabet. We observe that our watermark tolerates moderate substitution attacks.

**Additional discussion.** While our watermark is detectable in practice, and it exhibits an unremovability-quality tradeoff, the strong asymptotic tradeoff in Section 5 is somewhat dampened by the concrete parameters we consider. For example, the false negative rate is exponential in $-\sigma^4 T$, where $T$ is the number of distinct tokens. Although asymptotically this expression is negligible in the number of distinct tokens, for concrete values of $\sigma$ this term dominates. For example, suppose that the watermark was applied with $\sigma = 0.3$. We show that the false negative rate is at most $\exp(-(0.3)^4 T)$, where $T$ is the number of distinct tokens; to get any meaningful false positive guarantee we must have $T \geq (0.3)^{-4} \approx 125$. The vast majority of responses generated in our experiments had fewer than 100 distinct tokens, because we aimed to measure the detectability of paragraph-length texts of $\sim 300$ tokens.

**Quality.** In Figure 6, we plot quality scores for text generated by Qwen3.5-4B, using our watermark with various parameters of sigma. We compute these quality scores by asking Mistral-Instruct-7B to provide a score out of 100. We report the average scores across 12 responses per value of sigma, including responses of the three types we consider (essay, story, and code).

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

## A  Concentration bounds

Here, we use a standard Gaussian vector of dimension $n$ to mean a vector that whose components are independent Gaussians with unit variance.

**Fact 2** (Fernandez-Granda Lecture Notes, Theorem 2.7). *Let $\vec{x}$ be an iid standard Gaussian random vector of dimension $n$. For any $c \in (0,1)$, we have*

$$\Pr\left[n(1-c) < \|\vec{x}\|_2^2 < n(1+c)\right] \geq 1 - 2\exp\left(-\frac{nc^2}{8}\right).$$

**Fact 3** (Gaussian tail bound). *Let $X \sim \mathcal{N}(0, \sigma^2)$. For any $t \geq 0$,*

$$\Pr[|X| \geq t] \leq 2\exp\left(\frac{-t^2}{2\sigma^2}\right).$$

**Fact 4** (Hoeffding's Inequality). *Let $X_1, \ldots, X_k$ be independent random variables in $[a, b]$, and let $X = \sum_{i=1}^{k} X_i$. Then for any constant $\delta \geq 0$,*

$$\Pr\left[|X - \mathbb{E}[X]| \geq \delta\right] \leq 2\exp\left(\frac{-\delta^2}{\sum_{i=1}^{k}(b-a)^2}\right).$$

## B  Language models

A *language model* operates over a token alphabet $\mathcal{T} = \{t_1, \ldots, t_n\}$. We sometimes identify tokens $t_i$ with their indices $i$. We consider autoregressive language models that apply the softmax function to obtain a probability distribution over the next token. That is, given a prompt and tokens output so far, the model computes a probability distribution $p = [p_1, \ldots, p_n]$ over the next token as follows. The last layer of the model computes logits $\ell_i, \ldots, \ell_n$ for each token. Each logit is computed as

$$\ell_i = w_{i,0} + \sum_{j=1}^{m} w_{i,j} v_{i,j}$$

where $w_{i,0}$ is the *bias* and $\sum_{j=1}^{m} w_{i,j} v_{i,j}$ is a weighted average of that node's inputs. The probability $p_i$ is computed by applying the softmax function to the logits; that is,

$$p_i = \frac{e^{\ell_i}}{\sum_{j=1}^{n} e^{\ell_j}}.$$

Because the probabilities are computed using the softmax function, no probabilities are exactly equal to zero; rather, they are extremely small. However, in practice these extremely small probabilities are functionally zero. We formally treat these small probabilities as cryptographically negligible.

## C  Deferred proofs

### C.1  Proof of Theorem 1 (unremovability)

*Proof.* In the unremovability game, let $\vec{w}^*$ denote the original content and $\vec{w}_{\mathsf{wat}}$ denote the watermarked content. By our assumption about the adversary's posterior distribution over $\vec{w}^*$, we have that from the adversary's perspective, $\vec{w}^* \sim \mathcal{N}(0, \sigma^2 I)$. We will show that for any $z$ produced by the adversary, either $z$ is far from $\vec{w}^*$, or $z$ is watermarked with high probability.

For any fixed $z \in \mathbb{R}^n$, letting $v = \vec{w}^* - \vec{w}_{\mathsf{wat}} \sim \mathcal{N}(0, \sigma^2 I)$ and $u = z - \vec{w}_{\mathsf{wat}}$,

$$(z - \vec{w}^*) \cdot (\vec{w}_{\mathsf{wat}} - \vec{w}^*) = (v - u) \cdot v.$$

We now argue that the quantity $(v - u) \cdot v$ is likely to be large. Let $v_1$ denote the first component of $v$, let $\delta = \frac{1}{\sqrt{n}} \|u\|_2$, and let $f > 1$ be a constant.

$$
\begin{aligned}
\Pr_{v \leftarrow \mathcal{N}(0, \sigma^2 I)} [(v - u) \cdot v < \tau \sigma^2 n] &= \Pr_{v \leftarrow \mathcal{N}(0, \sigma^2 I)} \left[ \|v\|_2^2 - \delta v_1 \sqrt{n} < \tau \sigma^2 n \right] \\
&\leq \Pr_{v \leftarrow \mathcal{N}(0, \sigma^2 I)} \left[ \|v\|_2^2 - \delta \sigma f \sqrt{n} < \tau \sigma^2 n \text{ and } v_1 \leq \sigma f \right] + \Pr_{v \leftarrow \mathcal{N}(0, \sigma^2 I)} [v_1 > \sigma f] \\
&\leq \Pr_{v \leftarrow \mathcal{N}(0, \sigma^2 I)} \left[ \|v\|_2^2 - \delta \sigma f \sqrt{n} < \tau \sigma^2 n \right] + \frac{1}{2\pi} \exp\left( -f^2/2 \right) \\
&= \Pr \left[ \sigma^2 \|\mathcal{N}(0, I)\|_2^2 < \tau \sigma^2 n + \delta \sigma f \sqrt{n} \right] + \frac{1}{2\pi} \exp\left( -f^2/2 \right) \\
&= \Pr \left[ \|\mathcal{N}(0, I)\|_2^2 < \left( \tau + \frac{\delta f}{\sigma \sqrt{n}} \right) n \right] + \frac{1}{2\pi} \exp\left( -f^2/2 \right) \\
&\leq 2 \exp\left( -n \left[ 1 - \tau - \frac{\delta f}{\sigma \sqrt{n}} \right]^2 / 8 \right) + \frac{1}{2\pi} \exp\left( -f^2/2 \right) \qquad (1)
\end{aligned}
$$

Above, we made use of the facts that $v \sim \mathcal{N}(0, \sigma^2 I) \sim \sigma \mathcal{N}(0, I)$, and $v_1 \sim \mathcal{N}(0, \sigma^2)$. We also made use of the fact that for all $\ell \in \mathbb{R}$ and $u \in \mathbb{R}^n$,

$$
\Pr_{v \leftarrow \mathcal{N}(0, \sigma^2 I)} [u \cdot v = \ell] = \Pr_{v_1 \leftarrow \mathcal{N}(0, \sigma^2)} [v_1 \|u\|_2 = \ell],
$$

which follows from spherical symmetry of a Gaussian.

We've now upper bounded the probability that $(v - u) \cdot v$ is large in Equation (1); it remains to analyze this probability bound. Observe that for any $f = \omega(\sqrt{\log n})$, in Equation (1), $\frac{1}{2\pi} e^{-f^2/2}$ is negligible. For any $\delta = o(\sigma n / f) = o(\sigma n / \sqrt{\log n})$, in Equation (1) we have that $\tau n - \frac{\delta f \sqrt{n}}{\sigma}$ is at most $(1 - c)n$ for some constant $c = \tau + o(1)$. By Fact 2, $\Pr \left[ \|\mathcal{N}(0, I)\|_2^2 \leq \tau n - \frac{\delta f \sqrt{n}}{\sigma} \right] \leq \mathsf{negl}(n)$. Therefore, the expression in Equation (1) is negligible in $n$.

Applying the above, if $z$ is unwatermarked with non-negligible probability we must have $\|z - \vec{w}_{\mathsf{wat}}\|_2 \geq \omega \left( \frac{\sigma n}{\sqrt{\log n}} \right)$. By Fact 2, for any constant $\alpha$, $\Pr \left[ \|\vec{w}_{\mathsf{wat}} - \vec{w}^*\|_2 \geq \frac{\alpha \sigma n}{\sqrt{\log n}} \right] \leq \mathsf{negl}(n)$. By a union bound, if $z$ is unwatermarked then with overwhelming probability we must have $\|z - \vec{w}_{\mathsf{wat}}\|_2 \geq \omega \left( \frac{\sigma n}{\sqrt{\log n}} \right)$ and $\|\vec{w}_{\mathsf{wat}} - \vec{w}^*\|_2 < \frac{\alpha \sigma n}{\sqrt{\log n}}$. By a triangle inequality,

$$
\begin{aligned}
\|z - \vec{w}^*\|_2 &\geq \|z - \vec{w}_{\mathsf{wat}}\|_2 - \|\vec{w}_{\mathsf{wat}} - \vec{w}^*\|_2 \\
&\geq \omega \left( \frac{\sigma n}{\sqrt{\log n}} \right) - \frac{\alpha \sigma n}{\sqrt{\log n}} \\
&\geq \omega \left( \frac{\sigma n}{\sqrt{\log n}} \right).
\end{aligned}
$$

Finally, again by Fact 2, with overwhelming probability $\|\vec{w}_{\mathsf{wat}} - \vec{w}^*\|_2 = \Theta(\sigma \sqrt{n})$. Therefore, $\|z - \vec{w}^*\|_2 \geq \omega \left( \frac{\sqrt{n}}{\sqrt{\log n}} \right)$.

$\square$

## C.2    Proof of Theorem 2 (soundness of weight detector)

*Proof.* Let $x' \in \mathbb{R}^n$. We will show that with overwhelming probability, $\|x' - x_{\mathsf{wat}}\| > \frac{1}{2}\sigma n$. Since the detector outputs $\mathsf{true}$ only if $\|x' - x_{\mathsf{wat}}\| > \frac{1}{2}\sigma n$, this is sufficient to show that any fixed $x'$ will not be falsely detected.

Let $\Delta = x_{\mathsf{wat}} - x$, and observe that

$$
\begin{aligned}
\|x' - x_{\mathsf{wat}}\|^2 &= \|x' - (x + \Delta)\|^2 \\
&= \|\Delta - (x - x')\|^2 \\
&= \|\Delta\|^2 - 2\Delta \cdot (x - x') + \|x - x'\|^2
\end{aligned}
$$

Recall that by spherical symmetry of a Gaussian, $\Delta \cdot (x - x')$ is distributed as $\Delta_1 \|x - x'\|$ where $\Delta_1 \sim \mathcal{N}(0, \sigma^2)$. Therefore, with overwhelming probability, $2\Delta(x - x') \leq 2\sqrt{n}\|x - x'\|$. By Fact 2 and a union bound, with overwhelming probability we also have $\|\Delta\|^2 \geq 0.9\sigma^2 n^2$. Therefore,

$$
\|x' - x_{\mathsf{wat}}\|^2 \geq 0.9\sigma^2 n^2 + \|x - x'\|(\|x - x'\| - 2\sqrt{n})
$$

We now consider two cases. If $\|x - x'\| \geq 2\sqrt{n}n$, we have $\|x' - x_{\mathsf{wat}}\|^2 \geq 0.9\sigma^2 n^2$; therefore, $\|x' - x_{\mathsf{wat}}\| \geq \frac{1}{2}\sigma n$ as desired. If $\|x - x'\| < 2\sqrt{n}$, we have $\|x' - x_{\mathsf{wat}}\|^2 \geq 0.9\sigma^2 n^2 + 4n \geq 0.8\sigma^2 n^2$ for sufficiently large $n$. $\qquad\square$

## C.3 Proof of Lemma 1

*Proof.* We first show that Assumption 1 implies a useful relationship between $(z_t - \vec{w}_t^*)$ and the ratio between $p_t^i$ and $q_t^i$. Observe that

$$
\begin{aligned}
p_t^i &= \frac{e^{\ell_t^i + z_t}}{\sum_{r \in [n]} e^{\ell_r^i + z_r}} \\
&= \frac{e^{\ell_t^i + \vec{w}_t^* + (z_t - \vec{w}_t^*)}}{\sum_{r \in [n]} e^{\ell_r^i + z_r}} \\
&\approx \frac{e^{\ell_t^i + \vec{w}_t^* + (z_t - \vec{w}_t^*)}}{\sum_{r \in [n]} e^{\ell_r^i + \vec{w}_r^*}} \quad \text{by Assumption 1} \\
&= \frac{e^{\ell_t^i + \vec{w}_t^*}}{\sum_{r \in [n]} e^{\ell_r^i + \vec{w}_r^*}} \cdot e^{(z_t - \vec{w}_t^*)} \\
&= q_t^i \cdot e^{(z_t - \vec{w}_t^*)} \\
&\approx q_t^i + q_t^i \cdot (z_t - \vec{w}_t^*) \quad \text{using the approximation that } e^x \approx 1 + x.
\end{aligned}
$$

Rearranging, we have the useful fact:

$$
(z_t - \vec{w}_t^*) \approx p_t^i / q_t^i - 1
$$

where $\approx$ comes from Assumption 1 and the approximation $e^x \approx 1 + x$. Thus we have arrived at Assumption 2.

Armed with Assumptions 1 and 2, and Fact 1, we can prove the lemma. Our goal is to rewrite the inner product in terms of the expectation of $(\vec{w}_{\mathsf{wat}} - \vec{w}^*)_r$ for tokens $r$ appearing in the given text. Let $T$ denote the set of tokens over which $p^i$ has non-negligible probability. Let $p^i|_T$ denote the vector of probabilities when $p^i|_T$ is restricted only to tokens in $T$. Let $(z - \vec{w}^*)|_T$ and $(\vec{w}_{\mathsf{wat}} - \vec{w}^*)|_T$ denote the vectors $(z - \vec{w}^*)$ and $(\vec{w}_{\mathsf{wat}} - \vec{w}^*)$ restricted to the tokens in $T$.

We first manipulate the expression for the dot product and apply Assumption 2:

$$
\begin{aligned}
(z - \vec{w}^*)|_T \cdot (\vec{w}_{\mathsf{wat}} - \vec{w}^*)|_T &= \sum_{t \in T} (z_t - \vec{w}_t^*) \cdot (\vec{w}_{\mathsf{wat}} - \vec{w}^*)_t \\
&\leq \sum_{t \in T} (p_t^i / q_t^i - 1) \cdot (\vec{w}_{\mathsf{wat}} - \vec{w}^*)_t + \eta |(\vec{w}_{\mathsf{wat}} - \vec{w}^*)_t| \quad \text{by Assumption 2} \\
&= \mathbb{E}_{r \sim p^i|_T} \left[ \frac{1}{q_r^i} (\vec{w}_{\mathsf{wat}} - \vec{w}^*)_r \right] - \sum_{t \in T} (\vec{w}_{\mathsf{wat}} - \vec{w}^*)_t + \eta |(\vec{w}_{\mathsf{wat}} - \vec{w}^*)_t| \quad (2)
\end{aligned}
$$

We now analyze the expression $\mathbb{E}_{r \sim p^i|_T}\left[\frac{1}{q_r^i}(\vec{w}_{\mathsf{wat}} - \vec{w}^*)_r\right]$ to remove the term $\frac{1}{q_r^i}$, which cannot be determined from just the text. The high entropy and quality of $p^i$ lets us do exactly this, invoking Fact 1.

Recall that Fact 1 says that for the set of tokens $T$ over which $p^i$ has non-negligible probability, for all $t \in p^i$ we have $1/q_t^i \le c_1|T|/c_2$. Let $p^i|_T$ denote $p^i$ restricted to tokens in $T$ and renormalized. Applying this inequality, we have

$$
\begin{aligned}
\mathbb{E}_{r \sim p^i|_T}\left[\frac{1}{q_r^i}(\vec{w}_{\mathsf{wat}} - \vec{w}^*)_r\right] &= \sum_{t \in T} p_t^i(1/q_t^i)(\vec{w}_{\mathsf{wat}} - \vec{w}^*)_t \\
&\le \frac{c_1|T|}{c_2}\sum_{t \in T} p_t^i(\vec{w}_{\mathsf{wat}} - \vec{w}^*)_t \text{ by Fact 1} \\
&= \frac{c_1|T|}{c_2}\mathbb{E}_{r \sim p^i|_T}\left[(\vec{w}_{\mathsf{wat}} - \vec{w}^*)_r\right]
\end{aligned}
$$

Therefore,

$$
\begin{aligned}
\mathbb{E}_{r \sim p^i|_T}\left[(\vec{w}_{\mathsf{wat}} - \vec{w}^*)_r\right] &\ge \frac{c_2}{c_1|T|}\mathbb{E}_{r \sim p^i|_T}\left[\frac{1}{q_r^i}(\vec{w}_{\mathsf{wat}} - \vec{w}^*)_r\right] \\
&\ge \frac{c_2}{c_1|T|}\left((z - \vec{w}^*)|_T \cdot (\vec{w}_{\mathsf{wat}} - \vec{w}^*)|_T + \sum_{t \in T}(\vec{w}_{\mathsf{wat}} - \vec{w}^*)_t\right) - \eta|(\vec{w}_{\mathsf{wat}} - \vec{w}^*)_t| \\
&\text{by Equation (2)}
\end{aligned}
$$

which is a lower bound on this expected value with no dependence on $q$, as desired.

Finally, we analyze the expected value over the choice of $\vec{w}^*$. Recall that from the adversary's perspective, $\vec{w}^* \sim \mathcal{N}(0, \sigma^2 I)$. This immediately lets us simplify the term with $\eta$, using the mean of the folded normal distribution:

$$
\mathbb{E}_{\vec{w}^*}\left[\eta|(\vec{w}_{\mathsf{wat}} - \vec{w}^*)_t|\right] = \eta\sigma\sqrt{2/\pi}. \tag{3}
$$

As in the proof of Theorem 1, let $v = \vec{w}^* - \vec{w}_{\mathsf{wat}}$ and let $u = z - \vec{w}_{\mathsf{wat}}$. Then $(z - \vec{w}^*) \cdot (\vec{w}_{\mathsf{wat}} - \vec{w}^*) = (u - v) \cdot v$, where $v \sim \mathcal{N}(0, \sigma^2 I)$.

$$\mathbb{E}_{\vec{w}^*}\mathbb{E}_{r\sim p^i|_T}\left[(\vec{w}_{\mathsf{wat}}-\vec{w}^*)_r\right] \geq \mathbb{E}_{\vec{w}^*}\left[\frac{c_2}{c_1|T|}\left((z-\vec{w}^*)|_T\cdot(\vec{w}_{\mathsf{wat}}-\vec{w}^*)|_T+\sum_{t\in T}(\vec{w}_{\mathsf{wat}}-\vec{w}^*)_t-\eta|(\vec{w}_{\mathsf{wat}}-\vec{w}^*)_t|\right)\right]$$

$$= \mathbb{E}_{\vec{w}^*}\left[\frac{c_2}{c_1|T|}\left((z-\vec{w}^*)|_T\cdot(\vec{w}_{\mathsf{wat}}-\vec{w}^*)|_T+\sum_{t\in T}(\vec{w}_{\mathsf{wat}}-\vec{w}^*)_t\right)\right]-\frac{c_2\eta\sigma\sqrt{2/\pi}}{c_1}$$

$$\text{by Equation (3)}$$

$$= \mathbb{E}_{\vec{w}^*}\left[\frac{c_2}{c_1|T|}(z-\vec{w}^*)|_T\cdot(\vec{w}_{\mathsf{wat}}-\vec{w}^*)|_T\right]-\frac{c_2\eta\sigma\sqrt{2/\pi}}{c_1}$$

$$\text{since each }(\vec{w}_{\mathsf{wat}}-\vec{w}^*)_t\text{ has mean }0$$

$$= \mathbb{E}_{v\sim\mathcal{N}(0,\sigma^2 I)}\left[\frac{c_2}{c_1|T|}(u-v)|_T\cdot v|_T\right]-\frac{c_2\eta\sigma\sqrt{2/\pi}}{c_1}$$

$$= \mathbb{E}_{v\sim\mathcal{N}(0,\sigma^2 I)}\left[\frac{c_2}{c_1|T|}\left(\|v|_T\|_2^2-u|_T\cdot v|_T\right)\right]-\frac{c_2\eta\sigma\sqrt{2/\pi}}{c_1}$$

$$= \mathbb{E}_{v\sim\mathcal{N}(0,\sigma^2 I)}\left[\frac{c_2}{c_1|T|}\|v|_T\|_2^2\right]-\frac{c_2\eta\sigma\sqrt{2/\pi}}{c_1}\text{ since each }u_i\cdot v_i\text{ has mean }0$$

$$= c_2\sigma^2/c_1-\frac{c_2\eta\sigma\sqrt{2/\pi}}{c_1}$$

Therefore,

$$\mathbb{E}_{\substack{\vec{w}^*\\r\sim p^i|_T}}[\vec{w}_{\mathsf{wat}}-\vec{w}^*] \geq c_2\sigma^2/c_1-\frac{c_2\eta\sigma\sqrt{2/\pi}}{c_1}.$$

$\square$

### C.4 Proof of Theorem 3 (soundness of text detctor)

*Proof.* First, observe that $\mathbb{E}_{x_i\sim p_i}_{\vec{w}^*}[(\vec{w}_{\mathsf{wat}}-\vec{w}^*)_{x_i}] \geq 0$ for any distribution $p_i$ produced by the adversary. We've shown in Lemma 1 that for $p_i$'s that are $c_1$-high min-entropy and $c_2$-high quality, $\mathbb{E}_{x_i\sim p_i|_T}_{\vec{w}^*}[(\vec{w}_{\mathsf{wat}}-\vec{w}^*)_{x_i}] \geq c_2\sigma^2/c_1$, where $T$ is the set of tokens with non-negligible probability. Formally, we model the tokens appearing in the text as sampled from $p^i|_T$ rather than $p^i$, since the negligible probability tokens will never be sampled.

By assumption, at least a $\delta$ fraction of $p_i$'s have are $c_1$-high min-entropy and $c_2$-high quality. Therefore, letting $U$ denote the set of indices of distinct tokens in the response,

$$\frac{1}{|U|}\mathbb{E}\left[\sum_{i\in U}(\vec{w}_{\mathsf{wat}}-\vec{w}^*)_{x_i}\right] \geq \frac{\delta c_2\sigma^2}{c_1}-\alpha = 2\sigma^2\tau_{\mathsf{text}}$$

By Assumption 3, the $(\vec{w}_{\mathsf{wat}}-\vec{w}^*)_{x_i}$'s are independent. Furthermore, all $(\vec{w}_{\mathsf{wat}}-\vec{w}^*)_{x_i}$'s lie in $[-\lambda^{1/4},\lambda^{1/4}]$ with overwhelming probability by a standard Gaussian tail bound (Fact 3):

$$\Pr\left[\exists x_j\in U\text{ s.t. }\Delta(x_j)\notin[-\lambda^{1/4},\lambda^{1/4}]\right] \leq 2|U|\exp\left(\frac{-\lambda^{1/2}}{2\sigma^2}\right).$$

Let $X = \sum_{i \in U} (\vec{w}_{\mathsf{wat}} - \vec{w}^*)_{x_i}$. By a Hoeffding bound (Fact 4),

$$\Pr\left[|X - \mathbb{E}[X]| \geq \mathbb{E}[X]/2\right] \leq 2\exp\left(\frac{-2(\mathbb{E}[X]/2)^2}{|U|(2\lambda^{1/4})^2}\right)$$

$$\leq 2\exp\left(\frac{-2|U|^2\sigma^4\tau_{\text{text}}^2}{4|U|\sqrt{\lambda}}\right) \text{ since } \mathbb{E}[X] \geq |U|2\sigma^2\tau_{\text{text}}$$

$$= 2\exp\left(\frac{-|U|\sigma^4\tau_{\text{text}}^2}{2\sqrt{\lambda}}\right)$$

which is negligible in $\lambda$, as $|U| \geq \lambda$. □

### C.5 Proof of Theorem 4

*Proof.* Consider a given text, and recall that our detector only looks at distinct tokens. Therefore, let $x_1, \ldots, x_k$ denote the given text with the duplicate tokens removed. Observe that for each $i \geq \lambda$, the count during that iteration is $\sum_{j=1}^{i} \Delta(x_j)$, where each $\Delta(x_j)$ is an i.i.d. Gaussian $\mathcal{N}(0, \sigma^2)$. Therefore, their sum is also a Gaussian random variable $X \sim \mathcal{N}(0, i\sigma)$.

By Fact 3,

$$\Pr\left[X \geq i\tau_{\text{text}}\sigma^2\right] \leq 2\exp\left(\frac{-2i^2\tau_{\text{text}}^2\sigma^4}{2i\sigma^2}\right)$$

$$= 2\exp\left(-i\tau_{\text{text}}^2\sigma^2\right)$$

Since $i \geq \lambda$, this is negligible in $\lambda$. By a union bound, the probability that this sum exceeds the threshold for any $i$ is at most $k$ times this expression, which is also negligible. □

