# OpenReview forum: "Provably Robust Watermarks for Open-Source Language Models"
_TMLR — Under review for TMLR_

### Review · Reviewer_hkpv · 2026-04-04

**Summary Of Contributions:**

This paper proposes the first provable watermarking scheme for open-source LLMs. The authors propose to do watermarking by adding Gaussian perturbations to the bias of the last-layer of the language model. The authors provide formal definitions of unremovability in the open-source setting, prove that their watermarking satisfies these definitions under stated assumptions, and support their theoretical results with experiments on OPT models. The paper makes a meaningful first step toward provably robust open-source watermarking.

**Additional Comments:**

### Strengths

- **First formal formulation of watermarking in the open-source setups.** The paper is the first to define the provable watermarking scheme in open-source setups formally, with a clearly defined threat model and provable guarantees
- **The proposed approach is theoretically sound.** The inner product detector is interesting to read and shows advantages compared with prior sampling-based approaches. The unremovability-quality tradeoff is well-formulated, and the intuition is clear. Additionally, the current formulation is clean and can serve as a foundation for future work

### Weaknesses

- **[Major] The threat model is restricted.** The paper only considers attackers who can modify the last-layer biases of the model. The authors acknowledge this in Section 2.1, which argues that modifying other weights can compromise quality. However, in practice, an attacker could apply fine-tuning and quantization that go far beyond last-layer bias changes with affordable efforts. For example, parameter-efficient fine-tuning (LoRA or QLoRA) on a task-specific small dataset.
- **[Major] L2 distance is a poor proxy for model quality.** The paper uses L2 distance between weight vectors as the quality metric. However, open-source LLMs with different training recipes can have large L2 distances while having comparable quality. This means the whole framework based on L2 metric doesn't necessarily translate to real-world quality. Additionally, the experiments use Mistral-7B-Instruct as a judge. This creates a gap between the theoretical claim and the experimental setups.
- **[Major] Assumption 3 of Independence is a strong assumption with autoregressive LLMs.** The paper assumes the watermark perturbation signals are independent across the generated tokens. However, in autoregressive generation, the probability of each token is conditioned on the prior context. For example, if the context prefers a specific token, the bias impact becomes negligible.
- *[Minor] Experiments are placed in the appendix.* The experiment is important to justify the realism of the proposed theory. Therefore, it's suggested to move into the main paper.
- *[Minor] Outdated models are used in the experiment.* All experiments are conducted on OPT, which is relatively old and weak compared to the current state of the art. It is unclear whether the results would generalize to modern open-source models.

**Audience:**

Yes

**Audience Explanation:**

This paper addresses an important problem of open-source LLM watermarking. The formal framework, including the definitions of unremovability and the removability game, provides a valuable foundation for future work. This can be interesting to read for researchers working on GenAI watermarking.

**Claims And Evidence:**

Yes

**Claims Explanation:**

The claims are supported by the theoretical proof and experimental results. However, the problem space has a strong assumption and limitations. Please refer to Additional Comments for more details.

**Requested Changes:**

- Better justify the quality metric. The quality metrics are different in the theory (L2) and experiment (LLM-as-Judge). The authors should justify the connection between these metrics or explicitly acknowledge that the theoretical quality guarantee does not translate to a practical model quality guarantee and remains for future work.
- Justify independence assumption (Assumption 3) in the autoregressive LMs. Theorem 3 relies on the independence of watermark perturbation signals across distinct tokens (Assumption 3), which is a strong assumption for autoregressive LMs since the probability is strongly conditioned on prior content. The authors should provide either a formal justification for why this assumption holds approximately in the autoregressive LMs or provide an analysis of how violations of the independence assumption influence the detection rate.
- Move experiment to the main paper.

---

### Review · Reviewer_p83q · 2026-04-14

**Summary Of Contributions:**

# summary:

This paper studies watermarking for open-source language models, where the model weights and code are public. The authors argue that many prior LLM watermarking methods are sampler-based and therefore become ineffective once an adversary can directly inspect and modify the sampling code. To address this, the paper proposes embedding the watermark directly into the model weights by adding Gaussian perturbations to the last-layer bias terms, and then detecting the watermark either from model weights or from generated text. The main technical claim is that, under explicitly stated assumptions about the adversary’s uncertainty and the text distribution, an adversary cannot remove the watermark without making sufficiently large changes that significantly degrade model quality; moreover, the watermark remains detectable from text with enough entropy and enough distinct tokens. The paper also provides a formal removability game, soundness / unremovability definitions, and a theoretical analysis connecting parameter-space watermark signal to text-level detectability. Experiments on OPT-1.3B and OPT-6.7B aim to validate the detectability–robustness–quality tradeoff in practice.

# strength:

- **The paper is careful in its formalization and proof structure.**
  The authors clearly distinguish the open-source setting from the standard black-box / query-only watermarking setting, and they introduce a relatively complete set of definitions around watermarking, quality loss, unremovability, and soundness. The proofs are also organized in a logical way: first showing that high-quality adversarially modified models must still retain watermark signal in weight space, and then connecting that signal to detectability from text. This level of formal care is a genuine strength.
- **The paper studies a reasonably interesting direction.**
  Moving from sampler-based watermarking to watermarking directly in model weights is a meaningful shift in perspective for open-source models. In that sense, the paper addresses a setting that is underexplored and arguably more realistic for modern open-weight LLMs than prior work that assumes inference-time secrecy.

# weakness:

- **The actual method is naive, and the algorithmic novelty feels limited.**
  At the implementation level, the proposal is basically to add Gaussian noise to the last-layer bias terms and use that perturbation vector as the watermark key. The paper itself explicitly describes the scheme as “fairly simple,” and in the related-work discussion it also acknowledges that the construction is close in spirit to Zhao et al. (2024): both approaches effectively bias token preferences, with the main difference being that this paper replaces a uniform biasing rule with Gaussian perturbations and uses a weighted detector rather than a simple counting-style detector. Because of this, the method can feel less like a fundamentally new watermarking mechanism and more like a reformulation of token biasing from the sampler into the model’s final-layer biases. This makes the main novelty lie much more in the formalization and proof framework than in the watermark construction itself.
- **The core conclusion is not very surprising.**
  The paper essentially shows that once one injects a reproducible noise pattern into the model parameters, the resulting model carries a detectable signature; and if an adversary wants to erase that signature without knowing the original model exactly, it must perturb the model further, which then hurts quality more. This conclusion is not especially counterintuitive. In other words, the paper proves something that is reasonable and useful to formalize, but the final takeaway itself feels somewhat ordinary.
- **The experiments are not strong enough to fully support the broader claims.**
  The empirical section is relatively thin: it focuses on two OPT models, short generations of around 300 tokens, and a limited class of perturbation-style attacks as shown in Appendix.D. The quality evaluation is also indirect, relying on Mistral-7B-Instruct as a judge. The paper itself notes that the experiments are meant to demonstrate feasibility rather than optimality, and it also admits that the asymptotic theoretical tradeoff is weakened in the concrete parameter regimes used in practice. Overall, the experiments feel more like a sanity check than a convincing empirical validation.

**Audience:**

Yes

**Audience Explanation:**

Water marking is a good topic and the paper is rigorous.

**Broader Impact Concerns:**

No additional statement needed

**Claims And Evidence:**

Yes

**Claims Explanation:**

The claims are mostly written in a mathematical way, which is rigorous and solid.

**Requested Changes:**

See weakness in the first box. The author can modified the paper according to the weakness.

---

### Review · Reviewer_59Sv · 2026-05-17

**Summary Of Contributions:**

The authors propose a method for "watermarking" an open-source (open-weights + code) decoder language model by adding an isotropic Gaussian noise to the biases of the un-embedding matrix. Without knowing the exact noise vector drawn for watermarking, the modified model's weights can be published openly but the generated text outputs of the watermarked model can (according to the authors) provably be detected, under certain specific conditions.

Overall I found the paper very interesting and novel in the setup, being different from other watermarking techniques not working in the open-weight/open-code setup.

The paper tries to motivate the design of the algorithms in its first part, which makes it accessible without the math-heavy parts. On the other hand, I found the math a bit excessive and detached from the main flow. I'll give concrete suggestions below.

The biggest weakness of the paper is the experimental evaluation (and possibly the unclear robustness to post-edits). First, the utility of the generated text was only outsourced to LLM-as-a-judge with Mistral simply prompting "what is the quality of the generated text". This is, in my view, too coarse-grained and does not really say under which circumstances is the generated output useful or not. The authors mention some hacks to prevent repetitive phrases, which is a clear signal of an LLM "collapse". How about testing the model on some actual down-stream tasks, like text classification, summarization, or machine-translation? Second (and maybe I missed that), how robust is the watermarking to post-edits of the generated text? That is, can I prove it comes from the watermarked model after some manual edits of the generated text?

Also I'm not sure if the paper's releated work really capture the state of the art of early 2026. On page 6, you mention "Block et al. (2025) in a work released after ours" - so this is a little red flag to me, as this submission is submitted in April 2026 to TMLR. Does it mean this manuscript is from 2024 and is outdated now?

Section 2 (Technical overview) describes the setup in text, but reference later sections - how about putting everything in a linear story instead? For instance "Our scheme (Section 5)" - the reader is confused - should I read this section first, or go to Section 5 first, or what?

Now more detailed feedback:

* On page 2, you mention an adversary is given code C and produce code C'. What is code? I thought you only change the weights later, this is confusing and unspecified at this stage. (does it include just the inference code, training code, some weights, pre-training data, etc?)

* The added noise is $\mathcal{N}(0, \varepsilon^2)$; why don't you keep the standard notation $\sigma^2$ so it's clear in the entire paper what the parameter means? This would decrease the cognitive load; $\sigma^2$ is widely use default for Gaussians, $\varepsilon$ can be anything abstract.

* On page 4, "This region is essentially a halfspace" - this confused me. If you add an isotropic gaussian to the bias weight vector, you get a vector from a Gaussian sphere and its distance to the mean is governed by the tail bounds (which you also show in the appendix). So I don't get this "halfspace" intuition here. In the same paragraph later, "However, because the posterior distribution over ..." - I found this really confusing. Maybe a 2d sketch would help to demonstrate it? (but, of course, high-dimensional Gaussians are very weird and somehow unintuitive)

* Page 4, "Quality notion": You use Euclidean distance between model weights as a proxy for the quality of the watermarked model - but later on you ask Mistral to judge the quality of the output. I find this contradicting and confusing.

* Page 5: "We emphasize that our unremovability" - this can be removed, I felt it has been said before.

* Page 5: "We consider only models with last-layer biases" is imprecise. You test some transformer models later, so how about saying that you only "watermark bias terms in the unembedding matrix"? Then it's clear that their number is equal to the subword token inventory size.

* I like the recap of notation in Section 4. But you define $\mathbb{N}$ for positive non-zero integers first and then you define $X^*$ for $k \in \mathbb{Z}_{\geq 0}$ - does it mean you allow empty strings?

* I found section 4.1 extremely confusing and abstract. For example, "content distribution $\mathcal{C}$ of real vectors" - what is it? What does content mean - is it text content that is generated, is it neural net weights? Overall I'd suggest to be as less abstact here as possible; maybe explain what you do with using the bias vectors (because that's what you actually do), and leave out the generalization to "content vectors".

* Page 7: What is the "security parameter $\lambda$" again? Is it different from $\varepsilon$? And what is $1^{\lambda}$ - some identity function or what? This was confusing.

* Page 7: You clash the notation. In definition 1, you output $x'$ in "watermark" but later in Definition 3 you output $x'$ again in step 3. Very hard to follow.

* Page 7: What is the meaning of the "quality loss function"? It is defined but not really discussed; you write about some abstract "content" again. Too abstract.

* Page 7: Why is $\ell(\cdot)$ a parameter? It looks like some loss function to me. But it is from integers to reals -- why from integers? That's a bit confusing.

* Page 7: Using $\mathcal{A}$ for the adversary is an overkill - it does not benefit anything later on, as far as I can tell. Maybe simplify.

* Page 7: In Definition 4 the loss $\ell$'s domain is now $\mathbb{R}^n \times \mathbb{R}^n$ - which contradicts the definition before where it was from integers?!

* Page 7: You define "removability game" and then "Unremovability" and then talk about "unremovability game". Please keep it coherent.

* Page 8: In Algorithm 8 you introduce $\tau$ which has never been defined.

* Page 8: Algorithm 4: $\tau_{\mathrm{text}}$ undefined?

* Page 8: Therem 1 again uses "original content" etc., which is confusing terminology.

* Page 9: Why do you rebrand "entropy" as the reciprocal of each softmaxed probability? Entropy is an established concept for a probability distribution, and using it for something different just makes it more confusing.

* Page 9: Suggestions: How about using \exp( ) instaed of e^{} in the softmaxes? These formulas are quity tiny right now.

* Page 9: By the way - is the temperature parameter ommitted from the softmaxes by mistake or by choice? (also in appendix B). Because later you mention sampling with probability 0.9.

Disclaimer: I didn't check any of the math proofs in section C. Here I trust the authors.

**Audience:**

Yes

**Audience Explanation:**

Watermarking LLMs is an important topic

**Claims And Evidence:**

Yes

**Claims Explanation:**

See my comments above

**Requested Changes:**

If possible, simplify the abstract math concepts to the concrete case. A walk-through example of watermarking, generating, and detection would be extremely helpful to understand the mechanics beyond just abstract formulas and underspecified algorithms.

---

> ### Comment · Reviewer_59Sv · 2026-06-03
> **Thank your for your responses**
>
> Many thanks for the updates in the manuscript and your comments.
>
> Some details coulbe be further clarified:
>
> * The theoretical part assumes temperature 1.0 but the experiments use temperature 0.9. What does it mean?
>
> Minor:
>
> * Thanks for commenting on $\lambda$. As it is later used in Theorem 3 as a main pre-requisite ("The response contains at least $\lambda$ distinct tokens.") you might want to add $\lambda \in \mathbb{N}$ to Section 4 so it's super clear right away
> * "We watermark Qwen3.5-4B Qwen Team (2026)." -> change citet to citep

---

> > ### Author Response · Authors · 2026-06-04
> >
> > Re temperature: That is correct. One would expect the watermark to be at least as detectable in practice at a temperature of 1, compared to a temperate of 0.9. We will add an explicit note about this and did not intent to make a particular point with the temperature choice.
> >
> > Thanks, we'll make the further minor corrections.